# Which Multivariate Multi-Scale Entropy Algorithm Is More Suitable for Analyzing the EEG Characteristics of Mild Cognitive Impairment?

**DOI:** 10.3390/e25030396

**Published:** 2023-02-21

**Authors:** Jing Liu, Huibin Lu, Xiuru Zhang, Xiaoli Li, Lei Wang, Shimin Yin, Dong Cui

**Affiliations:** 1Hebei Key Laboratory of Information Transmission and Signal Processing, School of Information Science and Engineering, Yanshan University, Qinhuangdao 066004, China; 2National Key Laboratory of Cognitive Neuroscience and Learning, Beijing Normal University, Beijing 100875, China; 3Neurology Department, Chinese People’s Liberation Army Rocket Force Characteristic Medical Center, Beijing 100088, China

**Keywords:** complexity, correlation, multi-channel coupling, multivariate multi-scale entropy, mild cognitive impairment, electroencephalograph

## Abstract

So far, most articles using the multivariate multi-scale entropy algorithm mainly use algorithms to analyze the multivariable signal complexity without clearly describing what characteristics of signals these algorithms measure and what factors affect these algorithms. This paper analyzes six commonly used multivariate multi-scale entropy algorithms from a new perspective. It clarifies for the first time what characteristics of signals these algorithms measure and which factors affect them. It also studies which algorithm is more suitable for analyzing mild cognitive impairment (MCI) electroencephalograph (EEG) signals. The simulation results show that the multivariate multi-scale sample entropy (mvMSE), multivariate multi-scale fuzzy entropy (mvMFE), and refined composite multivariate multi-scale fuzzy entropy (RCmvMFE) algorithms can measure intra- and inter-channel correlation and multivariable signal complexity. In the joint analysis of coupling and complexity, they all decrease with the decrease in signal complexity and coupling strength, highlighting their advantages in processing related multi-channel signals, which is a discovery in the simulation. Among them, the RCmvMFE algorithm can better distinguish different complexity signals and correlations between channels. It also performs well in anti-noise and length analysis of multi-channel data simultaneously. Therefore, we use the RCmvMFE algorithm to analyze EEG signals from twenty subjects (eight control subjects and twelve MCI subjects). The results show that the MCI group had lower entropy than the control group on the short scale and the opposite on the long scale. Moreover, frontal entropy correlates significantly positively with the Montreal Cognitive Assessment score and Auditory Verbal Learning Test delayed recall score on the short scale.

## 1. Introduction

Epidemiological studies show that diabetes is a high-risk factor for age-related cognitive impairment and dementia. In particular, type 2 diabetes mellitus (T2DM) can lead to impaired cognitive function, increase the risk of mild cognitive impairment (MCI) and Alzheimer’s disease (AD), and seriously affect the survival and quality of life of patients [1]. MCI is the early stage of AD and mainly manifests in the decline of cognitive function, but daily living ability is not affected. However, with the development of the disease course, it not only evolves into dementia but also exists with patients for a long time, further affecting human health. Research data show that the prevalence of MCI in T2DM patients is as high as 45.0% [2]. With the continuous improvement of people’s living standards and the aging population’s deepening, the prevalence of T2DM has increased, and the number of patients with dementia caused by it has also increased. Therefore, early diagnosis, intervention, and treatment of MCI patients have great practical significance.

The electroencephalogram (EEG) is a nonlinear, non-stationary, and multi-dimensional complex signal. It has been widely used in researching epilepsy, AD, schizophrenia, MCI, and other brain diseases [3,4,5,6]. With the development of nonlinear dynamics, complexity analysis has become a popular trend in the study of EEG time series, which can reflect the characteristics of the dynamic system. Entropy represents the rate of new information generation and can be used to calculate the irregularity and complexity of nonlinear dynamic signals, which has been widely used in various fields [7,8,9].

Pincus proposed the concept of approximate entropy to measure the regularity and stability of time series [10] and used it in fetal heart rate analysis to detect subtle and potentially important heart rate differences that are not visually obvious [11]. Nie et al. evaluated human health by analyzing the approximate entropy of human pulse signals and found that in subjects with poor health, the individual adaptive ability would decrease, leading to a decrease in the approximate entropy value [12]. Richman and Moorman proposed sample entropy, which overcomes the shortcoming of approximate entropy not self-matching and reduces the dependence on data length [13]. In a study of pressure time series centers, Montesinos demonstrated that sample entropy is less dependent on the data length than approximate entropy, with greater consistency and ability to distinguish different experimental groups [14]. Based on the order pattern of elements in the time series, Bandt and Pompe proposed permutation entropy [15]. Permutation entropy has a simple operation process, fast calculation speed, and widespread use in various fields. Seker et al. analyzed the distribution of permutation entropy in five brain regions to observe the differences between MCI, AD, and the normal control group. They found that the entropy values of AD patients were significantly lower than that of the normal control group, and the MCI group was in the middle level [16]. However, permutation entropy is only a simple sort of multi-dimensional space sequence, ignoring the amplitude changes of elements in the sequence. Chen et al. defined the concept of fuzzy entropy for the first time. As an improvement of the sample entropy algorithm, fuzzy entropy blurs the similarity measurement formula with an exponential function, making entropy values transition smoothly with changing parameters. At the same time, it inherits sample entropy’s characteristics of relative consistency and the ability to handle short data sets [17]. Liu et al. proposed increment entropy, which obtained symbol information based on the fluctuation trend mapping of adjacent elements and amplitude information based on quantitative resolution and combined the two to quantify irregular time series. In the study of real EEG signals of epilepsy, increment entropy can well detect the amplitude and structural changes of time series and has a good performance in detecting epileptic seizures [18]. Dispersion entropy is a new irregularity index proposed by Rostaghi and Azami [19], which extracts signal features by mapping time series into finite integer time series. Rostaghi et al. used dispersion entropy for fault diagnosis and state detection of industrial rotating equipment. They found that dispersion entropy can detect changes in signal amplitude and frequency simultaneously, with better results than approximate entropy and permutation entropy. The calculation speed is faster than the approximate entropy [20].

Most of the actual physiological signals are multivariate signals. The above methods are only suitable for processing univariate time series and cannot express the long-term correlation and complexity of multivariate time series. Researchers extend the concept of entropy and combine it with multi-scale to propose multivariate multi-scale entropy to analyze multi-channel signals. Ahmed and Mandic presented multivariate multi-scale sample entropy (mvMSE) based on multi-scale and sample entropy and analyzed multi-channel signals on different scales [21]. This algorithm can analyze the complexity of multivariate time series and widespread use in physical and physiological signals [22,23,24]. Li et al. substituted the fuzzy membership function for the hard threshold criterion of pattern similarity judgment in mvMSE and obtained multivariate multi-scale fuzzy entropy (mvMFE). Simulation results showed that introducing the fuzzy membership function could effectively improve the statistical stability of the algorithm, and this algorithm could use to guide the research of noninvasive early warning of cardiovascular diseases [25]. Subsequently, Azami and Escudero proposed the refined composite multivariate multi-scale fuzzy entropy (RCmvMFE), which can use to analyze short time series and improve the stability of mvMFE [26]. It has been applied to rolling bearing fault diagnosis [27], horizontal oil-water two-phase flow analysis [28], multi-channel financial data dynamic complexity measurement [29], and other research fields. Morabito et al. proposed multivariate multi-scale permutation entropy (mvMPE) algorithm based on permutation entropy to evaluate the complexity of physiological signals. This algorithm has been used to distinguish the brain states of subjects with AD and MCI patients from those of normal healthy older people. This algorithm has the advantages of fast operation speed, simple concept, and robustness to noise and artifacts.

However, the mvMPE has the same defect as permutation entropy in ignoring the difference in amplitude [30]. Azami et al. introduced multivariate multi-scale dispersion entropy (mvMDE) and used this algorithm to analyze 148-channel MEG signals to distinguish AD patients from controls. The results showed that: on the short scale, the mean value of mvMDE in AD patients was lower than the control group; on the long scale, the mean value of mvMDE in AD patients was greater than the control group, which was consistent with the results obtained by mvMFE algorithm [31]. Wang et al. extended the increment entropy to multi-scale increment entropy and multivariate multi-scale increment entropy (mvMIE). They used them to detect flow pattern transition in multi-phase flow. The results showed that mvMIE could effectively detect the evolution behavior of different flow patterns over time as flow conditions change and can reveal the dynamic complexity of varying flow patterns [32].

The entropy algorithm has widespread use in biology, finance, engineering, neuroscience, and other fields, and it has also made some achievements in analyzing EEG signals of patients with neurological diseases. Studies have shown that complexity and functional connectivity coexist in actual EEG signals, especially in MCI EEG signals [33,34]. However, so far, the articles using the multivariate multi-scale entropy algorithm mainly analyze the complexity of multi-channel data without explicitly describing what characteristics of signals these algorithms measure and what factors affect these characteristics.

In this article, we analyze six commonly used multivariate multi-scale entropy algorithms from a new perspective and discuss the combined effect of the complexity of each channel signal and the coupling strength between channels on the multivariable entropy value for the first time, which is unprecedented. This paper solves the problem that other reports do not specify what signal characteristics the algorithm measures and which factors affect these features. Moreover, it also studies which algorithm is more suitable for analyzing MCI EEG signals and uses it to analyze the actual EEG signal. Firstly, we simulate and analyze each algorithm using signals with different complexity and correlation to explain explicitly which characteristics each algorithm measures. After that, we jointly analyze the relationships of signal coupling strength, single-channel entropy, and multi-channel entropy to deeply understand the influence of signal complexity and coupling strength on multi-channel entropy. Secondly, we simulate the performance of each algorithm using signals with different noise intensities and data lengths. Consider the above simulation results to search for the algorithm suitable for analyzing MCI EEG signals. Then, we used the RCmvMFE algorithm to analyze the EEG signals of patients with MCI and normal cognitive to explore the nonlinear dynamic characteristics of the EEG signals of MCI. Finally, we analyzed the correlation between the RCmvMFE value of all participants and their neuropsychological scale scores to investigate whether there is a correlation between nonlinear dynamic characteristics of EEG signals and cognitive function.

This paper structure is as follows: the second section is the theoretical introduction of six kinds of multivariate multi-scale entropy algorithms and the acquisition of participants’ information and EEG signals; the third section includes the simulation analysis of multivariate multi-scale entropy algorithm and the actual EEG signal analysis; the fourth section is a discussion of this study and other related research results; and the fifth section is the conclusion of this paper.

## 2. Materials and Methods

The mvMSE algorithm combines multivariate sample entropy and multi-scale and can analyze multi-channel signals with different scale factors. The mvMFE algorithm uses a fuzzy membership function to replace the hard threshold criterion of pattern similarity judgment in mvMSE, which makes the entropy transition smoothly with parameter changes and effectively improves the statistical stability of the algorithm. The RCmvMFE algorithm improves in two aspects based on mvMFE, including enhancing the coarse-grained algorithm and introducing the adjustment factor to improve the fuzzy membership function to define the similarity. It avoids information loss when the data length of different scale factors is very diverse and can stably detect the dynamic behavior of complex systems. The mvMPE algorithm extends the permutation entropy algorithm to the multivariate time series. It only needs to consider whether the sequence pattern is the same and does not need to consider the threshold value and the distance between the space vectors. It has the advantages of a simple operation process and fast calculation speed. The mvMDE algorithm maps the time series into a finite integer time series to construct the dispersion patterns, which can simultaneously detect the change of signal amplitude and frequency and its effect better than the permutation entropy. The mvMIE algorithm produces incremental time series according to the fluctuation tendency of adjacent elements. It obtains the sign and the magnitude parts to quantize irregular time series in combination patterns, which can effectively detect the dynamic complexity of different time series.

### 2.1. Multivariate Multi-Scale Sample Entropy (mvMSE)

From a *p*-channel time series $U={\left\{u_{k,a}\right\}}_{a=1,2\ldots L}^{k=1,2\ldots p}$ with the data length *L*, we gain a coarse-grained time series $X={\left\{x_{k,b}\right\}}_{b=1,2\ldots N}^{k=1,2\ldots p}$ through the following coarse-grained process, as in Equation (1):(1)$$x_{k,b}{}^{\left(s\right)}=\frac{1}{s}\sum_{a=(b-1)s+1}^{bs}u_{k,a},1\leq k\leq p,1\leq b\leq \left\lfloor \frac{L}{s}\right\rfloor =N$$Here s stands for the scale factor.

The multivariate embedding vector $X_{m}(i)$ for coarse-grained time series X expresses as:(2)$$\begin{array}{ll} X_{m}(i)= & [x_{1,i},x_{1,i+d_{1}},\ldots ,x_{1,i+(m_{1}-1)d_{1}},\\ & x_{2,i},x_{2,i+d_{2}},\ldots ,x_{2,i+(m_{2}-1)d_{2}},\ldots ,\\ & x_{p,i},x_{p,i+d_{p}},\ldots ,x_{p,i+(m_{p}-1)d_{p}}] \end{array}$$
where each channel’s embedding dimension $m_{k}$ and time delay $d_{k}$ can be different values, for convenience, we set the same embedding dimension m and delay d for each channel and n=m×d, i=1,2,…,N−n.

Calculate the Chebyshev distance $d_{ij}^{m}$ between the multivariate embedding vector $X_{m}(i)$ and $X_{m}(j)$ by using the following formula:(3)$$d_{ij}^{m}=d[X_{m}(i),X_{m}(j)]=\max_{l=1,2\ldots m}(|x(i+l-1)-x(j+l-1)|),i\neq j$$

Given the similarity tolerance r, the probability of $d_{ij}\leq r,i\neq j$ is $\phi_{i}^{m}(r)=\frac{1}{N-n-1}P_{i}$, and then define a global quantity:(4)$$\phi^{m}(r)=\frac{1}{N-n}\sum_{i=1}^{N-n}\phi_{i}^{m}(r)$$

There are p ways to extend the embedding dimension from m to m+1, and the expansion process can express as $\left[m_{1},m_{2},\ldots ,m_{p}\right]$ to $\left[m_{1},\ldots ,m_{k}+1,\ldots ,m_{p}\right]$. Thus, we obtained p new multivariate embedding vectors $X_{m+1}(i)$. Repeat the above steps. Finally, we get the global quantity with embedding dimension m+1 is:(5)$$\phi^{m+1}(r)=\frac{1}{p\times (N-n)}\sum_{i=1}^{p\times (N-n)}\phi_{i}^{m+1}(r)$$

The multivariate multi-scale sample entropy defined by Shannon’s theorem is:(6)$$mvMSE(U,m,d,r)=-\ln(\phi^{m+1}(r)/\phi^{m}(r))$$

### 2.2. Multivariate Multi-Scale Fuzzy Entropy (mvMFE)

According to Equations (1)–(3), we get the Chebyshev distance $d_{ij}^{m}$ between $X_{m}(i)$ and $X_{m}(j)$. Introduce the fuzzy membership function is to define the similarity degree $D_{ij}^{m},i\neq j$ as follows:(7)$$D_{ij}^{m}(r)=\left\{\begin{array}{cc} 1 & ,\;0\leq d_{ij}^{m}\leq r\\ \exp(-\ln2{\left(\frac{d_{ij}^{m}-r}{r}\right)}^{2}) & ,d_{ij}^{m}>r \end{array}\right.$$

Define the global quantity with the embedding dimension m as:(8)$$\phi^{m}(r)=\frac{1}{N-n}\;\sum_{i=1}^{N-n}\left(\frac{1}{N-n-1}\sum_{j=1,j\neq i}^{N-n}D_{ij}^{m}\right)$$

We get p new multivariate embedding vectors $X_{m+1}(i)$ based on the extension method in mvMSE. Similarly, we defined the global quantity with embedding dimension m+1 as:(9)$$\phi^{m+1}(r)=\frac{1}{p\times (N-n)}\sum_{i=1}^{p\times (N-n)}\left(\frac{1}{p\times (N-n)-1}\sum_{j=1,j\neq i}^{p\times (N-n)}D_{ij}^{m+1}\right)$$

The multivariate multi-scale fuzzy entropy defined by Shannon’s theorem is:(10)$$mvMFE(U,m,d,r)=-\ln(\phi^{m+1}(r)/\phi^{m}(r))$$

### 2.3. Refined Composite Multivariate Multi-Scale Fuzzy Entropy (RCmvMFE)

The algorithm of RCmvMFE has made two improvements based on mvMFE, one of which is to improve the traditional coarse-grained process. We can get s different coarse-grained time series through the improved coarse-grained process. Repeat the process of mvMFE for each coarse-grained time series, and finally calculate the entropy after averaging the obtained s global quantities. The improved coarse-grained process is as follows:(11)$$x_{k,b,t}{}^{\left(s\right)}=\frac{1}{s}\sum_{a=(b-1)s+t}^{bs+t-1}u_{k,a},1\leq b\leq \left\lfloor \frac{L}{s}\right\rfloor =N,1\leq k\leq p,1\leq t\leq s$$

Based on Equations (2) and (3), we get s Chebyshev distance $d_{ij,t}^{m}$ of s multivariate embedding vectors. The second improvement is to introduce the improved fuzzy membership function to define the similar degree $D_{ij,t}^{m},i\neq j,1\leq t\leq s$ as follows:(12)$$D_{ij,t}^{m}(r)=\left\{\begin{array}{l} \begin{array}{cc} 1 & \;,\;0\leq d_{ij,t}^{m}\leq \lambda r \end{array}\\ \exp(-\ln2{\left(\frac{d_{ij,t}^{m}-\lambda r}{r}\right)}^{2})\begin{array}{cc} , & d_{ij,t}^{m}>\lambda r \end{array} \end{array}\right.$$

Here the adjustment factor λ ranges from 0.5 to 1.5 typically.

When the embedding dimension is m and m+1, the average of s global quantities express as follows:(13)$${\bar{\phi }}^{m}(r)=\frac{1}{s}\sum_{t=1}^{s}\frac{1}{N-n-1}\sum_{j=1,j\neq i}^{N-n}D_{ij,t}^{m}$$
(14)$${\bar{\phi }}^{m+1}(r)=\frac{1}{s}\sum_{t=1}^{s}\frac{1}{p\times (N-n)-1}\sum_{j=1,j\neq i}^{p\times (N-n)}D_{ij,t}^{m+1}$$

The refined composite multivariate multi-scale fuzzy entropy defined by Shannon’s theorem is:(15)$$RCmvMFE(U,m,d,r,\lambda )=-\ln({\bar{\phi }}^{m+1}(r)/{\bar{\phi }}^{m}(r))$$

### 2.4. Multivariate Multi-Scale Permutation Entropy (mvMPE)

Through Equation (1), we get the coarse-grained time series $X={\left\{x_{k,b}\right\}}_{b=1,2\ldots N}^{k=1,2\ldots p}$. By introducing the embedding dimension m and time delay d, we reconstruct the coarse-grained time series:(16)$$X_{m}(i)=\{x_{k,i},x_{k,i+d},\ldots ,x_{k,i+(m-1)d}\},1\leq i\leq N-(m-1)d$$

Sort the elements in each row of $X_{m}(i)$ and match them with m! possible patterns $\pi_{j},j=1,2\ldots m!$. Calculate the probability of each pattern as follows:(17)$$P(\pi_{j})=\frac{\#\{i|i\leq N-(m-1)d,type(X_{m}(i))=\pi_{j}\}}{(N-(m-1)d)\times p}$$

The multivariate multi-scale permutation entropy defined by Shannon’s theorem is:(18)$$mvMPE(U,m,d)=-\sum_{j=1}^{m!}P(\pi_{j})\ln(P(\pi_{j}))$$

### 2.5. Multivariate Multi-Scale Dispersion Entropy (mvMDE)

Through Equation (1), we get the coarse-grained time series $X={\left\{x_{k,b}\right\}}_{b=1,2\ldots N}^{k=1,2\ldots p}$, then maps to Y through the normal cumulative distribution function (NCDF):(19)$$y_{k,b}=\frac{1}{\sigma_{k}\sqrt{2\pi }}\int_{-\infty }^{x_{k,b}}\exp(-\frac{{\left(t-\mu_{k}\right)}^{2}}{2\sigma_{k}{}^{2}})dt$$

Here $\mu_{k}$ and $\sigma_{k}$ are the mean and the standard deviation of the time series X. The value of Y ranges from 0 to 1 and then mapping Y to Z, which ranges from 1 to c:(20)$$Z_{k,b}^{c}=round(c\cdot y_{k,b}+0.5)$$

A multivariate embedding vector $Z_{m}(i),1\leq i\leq N-(m-1)d$ is generated based on Equation (2). Set the embedding dimension and the time delay to the same value, expressed as $m_{k}=m$, $d_{k}=d$. Therefore, the length of each row vector in $Z_{m}(i)$ is $\sum_{k=1}^{p}m_{k}=m\times p$.

We are nonredundant to extract m elements from each row vector $Z_{m}(j)$, and there are $C_{m\times p}^{m}$ options. Defined each option as $\phi_{q}(j)$, $q=1,2,\ldots ,C_{m\times p}^{m}$, j=1,2,…,N−(m−1)d. Map the selected m elements into a dispersion pattern $\pi_{v_{0}v_{1}\ldots v_{m-1}}$. The following formula obtains the relative frequency:(21)$$p(\pi_{v_{0}v_{1}\ldots v_{m-1}})=\frac{\#\{j|j\leq N-(m-1)d,\phi_{q}(j)has\;type\;\pi_{v_{0}v_{1}\ldots v_{m-1}}\}}{(N-(m-1)d)\times C_{mp}^{m}}$$

The multivariate multi-scale dispersion entropy defined by Shannon’s theorem is:(22)$$mvMDE(U,m,d,c)=-\sum_{\pi =1}^{c^{m}}p(\pi_{v_{0}v_{1}\ldots v_{m-1}})\log(p(\pi_{v_{0}v_{1}\ldots v_{m-1}}))$$

### 2.6. Multivariate Multi-Scale Increment Entropy (mvMIE)

Through Equation (1), we get the coarse-grained time series $X={\left\{x_{k,b}\right\}}_{b=1,2\ldots N}^{k=1,2\ldots p}$, then calculate its increment time series $z_{k,b-1}=x_{k,b}-x_{k,b-1}$. The increment time series $Z={\left\{z_{k,b-1}\right\}}_{b=1,2\ldots N}^{k=1,2\ldots p}$ is reconstructed according to Equation (16) to obtain $Z_{m}(i)$, i=1,2,…,N−(m−1)d−1.

Each element of $Z_{m}(i)$ maps into a word contains two parts: the sign and the magnitude. This formula generates the sign part:(23)$$s_{k,i+l}=\left\{\begin{array}{c} +1,{\;z}_{k,i+l}>0\\ 0,{\;z}_{k,i+l}=0\\ -1,{\;z}_{k,i+l}<0 \end{array}\right.$$

The following formula generates the magnitude part:(24)$$q_{k,i+l}=\left\{\begin{array}{cc} 0, & std(Z_{m}(i))=0\\ \min(R,\left\lfloor \frac{\left|z_{k,i+l}\right|\times R}{std(Z_{m}(i))}\right\rfloor ), & std(Z_{m}(i))\neq 0 \end{array}\right.\;1\leq k\leq p,1\leq i\leq N-(m-1)d-1,1\leq l\leq (m-1)d$$

where R is the quantization resolution generally set R≤4. By formula $w_{k,i}=U_{l=0}^{(m-1)d}s_{k,i+l}q_{k,i+l},1\leq k\leq p,1\leq i\leq N-(m-1)d-1$, we combine the sign part and magnitude pate. There are up to ${\left(2\times (R+1)+1\right)}^{m}$ possible combination patterns for time series with embedding dimension m and quantization resolution R.

Define $w_{n}$ as each unique combination pattern in $w_{k,i}$, and count its number as $Q(w_{n})$. The relative frequency is defined as: (25)$$p(w_{n})=\frac{Q(w_{n})}{(N-(m-1)d-1)p}$$

The multivariate multi-scale increment entropy defined by Shannon’s theorem is:(26)$$mvMIE(U,m,d,R)=-\sum_{n=1}^{{\left(2(R+1)+1\right)}^{m}}p(w_{n})\log(p(w_{n}))$$

### 2.7. Participants and Neuropsychological Tests

The present study included 20 subjects with T2DM who enrolled in the Special Medical Center of the Chinese People’s Liberation Army Rocket Force, whose medical diagnosis met the World Health Organization requirements. All 20 subjects received a general demographic assessment and a series of standardized neuropsychological assessments. Such as the Mini-mental State Examination (MMSE), the Montreal Cognitive Assessment (MoCA), and the Auditory Verbal Learning Test (AVLT) to test subjects’ memory and recall, including AVLT immediate recall, AVLT delayed recall and AVLT long-delayed recognition. The Trail Test parts A, B, and the Wechsler Adult Intelligence Scale (WAIS) to test the subjects’ executive ability and attention, the Boston Naming Test and the Semantic Fluency Test to test the subjects’ language ability, and the Frequently Asked Questions (FAQ) to test the subjects’ ability of daily living. Divide all subjects into two groups (8 control subjects and 12 MCI subjects) according to the score of the neuropsychological scale. The grouping criteria met the inclusion and exclusion criteria of MCI [33]. The neuropsychological scale scores were consistent with normality and homogeneity of variance. Statistical analysis of the neuropsychological scale score using an independent sample *t*-test in the statistical analysis software SPSS (version 25.0). The statistical results are shown in Table 1 and expressed as mean ± standard deviation.

### 2.8. EEG Recording and Preprocessing

We collected the EEG signals in a quiet-dark room in the Neurology Department of the Rocket Force Special Medical Center of the Chinese People’s Liberation Army. Use the GES300 (Electrical Geodesics Inc., Eugene, Oregon, USA) 128-channel acquisition device to record the scalp EEG signals of participants. During EEG signal collection, participants sit in a comfortable armchair and stay in a closed-eye resting state. Bilateral mastoid processes served as reference electrodes. The impedance of all electrodes keeps below 5 KΩ, and the sampling frequency is 1000 Hz. Record the EEG signals with a duration of 300 s. We use a band-pass filter (0–200 Hz) and a notch filter (50 Hz) to filter the collected EEG signals. Afterward, we use the wavelet-enhanced independent component analysis method to preprocess the signal and remove EMG, EOG, and ECG artifacts. The wavelet-enhanced independent component analysis method describes in detail in Reference [35]. Then down-sample to 500 Hz and obtain the preprocessed 180 s EEG data. Select 32 electrodes distributed evenly across the scalp of the brain for analysis. These 32 electrode distributions meet the requirements of international standard leads and often use for study by researchers [36]. Divide them into six regions: frontal (F), central (C), parietal (P), occipital (O), left temporal (LT), and right temporal (RT). The schematic diagram of brain regions is shown in Figure 1.

## 3. Results

### 3.1. Simulation Analysis

In the simulation analysis of the characteristics of the following synthetic signals, the relevant parameters of the six multivariate multi-scale entropy algorithms are set as follows according to some related references [27,31,32]:

In mvMSE, mvMFE, and RCmvMFE algorithms, we set each channel’s embedding dimension and time delay to 2 and 1, respectively, and set the similarity tolerance to r=0.15×std(standard deviation). Set the adjustment value in RCmvMFE to λ=0.8.

In mvMDE, mvMPE, and mvMIE algorithms, we set each channel’s embedding dimension and time delay to 3 and 1, respectively. Set the number of classes in mvMDE to c=3 and the quantifying resolution in mvMIE to R=4.

#### 3.1.1. Complexity Analysis

To evaluate these algorithms’ ability to analyze the complexity of multivariate time series, we generated an uncorrelated three-channel time series consisting of 1/f noise and white Gaussian noise (WGN). The 1/f noise is a signal whose power spectral density is inversely proportional to frequency and has the characteristic of long-range correlation. The irregularity of 1/f noise is lower than that of WGN, but its complexity is higher than that of WGN. WGN is a signal with zero mean and one standard deviation. The total number of generated time series channels is always equal to 3, and as the number of 1/f noise channels decreases, the number of WGN channels increases correspondingly. Set the data length to 6000, and repeat the generation 20 times. Calculate the entropy values of the generated three-channel time series by six algorithms and compute their mean and standard deviation. The results are shown in Figure 2.

As seen from the figure, for the four algorithms of mvMSE, mvMDE, mvMFE, and RCmvMFE, the entropy value of the three-channel 1/f noise signal almost remains unchanged or slowly decreases with the scale factor’s increase, while the entropy values of the other three variables time series decrease monotonically. When the scale factor is bigger than or equal to 2, the entropy value increases gradually with the 1/f noise channel number and reaches the highest when the three channels are all 1/f noise. That is consistent with the fact that multivariate 1/f noise is structurally more complex than multivariate WGN. Among them, the mvMSE entropy curves of the first two kinds of signals show an aliasing phenomenon when the scale factor is greater than 15 and the differentiation ability is poor. The entropy curves of mvMDE and mvMFE have a poor ability to distinguish four kinds of signals when the scale factor is 1 and 2, respectively. However, RCmvMFE can distinguish the four signals well at any scale, so RCmvMFE has better signal discrimination ability than other algorithms. For the mvMIE algorithm, the entropy value gradually increases with the increase in the 1/f noise channel number at any scale factor. However, there are some overlaps between the entropy results at a large scale and distinguish the four three-channel signals not well. However, the performance of the mvMPE algorithm is contrary to the above five algorithms: the entropy is unchanged with the increase in the scale factor, and the entropy decreases with the rise of the 1/f noise channel number at any scale factor. Moreover, there are some overlaps between the entropy results at a large scale and distinguish the four three-channel signals not well. 

From the analysis results, we can conclude that the four algorithms of mvMSE, mvMDE, mvMFE, and RCmvMFE can use to analyze the intra-channel correlation and multivariate signal complexity. Among them, the RCmvMFE algorithm has the best signal discrimination ability. The mvMIE algorithm cannot reflect the intra-channel correlation and only be used to analyze the complexity of multivariate signals, but the signal discrimination is not good. The mvMPE algorithm only measures signal irregularity and cannot reflect intra-channel correlation.

#### 3.1.2. Correlation Analysis

To evaluate these algorithms’ ability to analyze the inter-channel correlation of time series, we generated the two-channel 1/f noise sequence and the two-channel WGN sequence, respectively, with inter-channel correlation and inter-channel uncorrelation. For both the two-channel sequence with inter-channel correlation, the correlation coefficient between channels is 0.95. The data length is 6000 and repeats the generation 20 times. Calculate the entropy values of the generated data by six algorithms and compute their mean and standard deviation. The results are shown in Figure 3.

As seen from the figure, for the three algorithms of mvMSE, mvMFE, and RCmvMFE, the entropy value of the correlated signals is always greater than the uncorrelated signals at any scale factor, whether it is a 1/f noise signal or WGN signal. That is consistent with the fundamental physics that the complexity of correlated signals is higher than the uncorrelated ones. The entropy values of the WGN signal decrease with the increase in the scale factor, while the entropy values of the 1/f noise signal basically remain unchanged. Finally, the correlated 1/f noise has the highest value, followed by the uncorrelated 1/f noise, correlated white noise, and uncorrelated white noise. The RCmvMFE algorithm shows this phenomenon when the scale factor is greater than 5, while the mvMSE and mvMFE algorithms show this phenomenon when the scale factor is greater than 6 and 15, respectively. These three algorithms reflect the intra-channel and the inter-channel correlation, and they can effectively distinguish the white noise and 1/f noise from the inter-channel correlation and uncorrelation on a large scale. Among them, with the increased scale factor, the RCmvMFE algorithm can distinguish the four signals earlier than the mvMSE and mvMFE algorithms and has better signal discrimination ability.

For the mvMDE algorithm, the entropy value of the 1/f noise signal remains unchanged with the increase in scale factor. The entropy value with correlated signals is always smaller than the uncorrelated signals and they are well distinguished at any scale factor. The entropy value of the two WGN signals with correlated and uncorrelated decreases with the scale factor’s increase. However, the entropy values of the two signals are too close to well distinguished. 

For the mvMPE algorithm, regardless of whether there is a channel correlation between 1/f noise and white noise signal, the entropy value basically remains unchanged with the scale factor’s increase, and the entropy value of the 1/f noise signal is always smaller than that of the white noise signal. The entropy values of correlated and uncorrelated signals are too close that it is impossible to distinguish them well, which is the case for both 1/f noise and white noise.

For the mvMIE algorithm, regardless of whether there is a channel correlation between 1/f noise and white noise signal, the entropy value decreases with the increase in scale factor, and the entropy value of the 1/f noise signal is always greater than that of the white noise signal. This algorithm can identify the 1/f noise signal and WGN signal, but it cannot distinguish whether there is a correlation between channels.

According to the above analysis, mvMSE, mvMFE, and RCmvMFE algorithms reflect the intra-channel correlation and can distinguish whether there is a correlation between signal channels. Among them, the RCmvMFE algorithm can distinguish four kinds of signals at a smaller scale. The MvMDE can only recognize whether there is a correlation between 1/f noise signal channels. However, mvMPE and mvMIE cannot use to analyze the correlation between channels.

#### 3.1.3. Joint Analysis of Coupling and Complexity

We adopt the coupled *MIX* model for joint analysis to further study the influence of multi-channel signals’ coupling strength and their single-channel complexity on multi-channel entropy. The *MIX* model comprises periodic and random noise signals in a particular proportion.
(27)MIX(p)=(1−p)×x+p×y

Here, the periodic signal is $x(t)=\sqrt{2}\sin(2\pi t/f_{s}),f_{s}=12\mathrm{Hz},t=0,1,\ldots ,N-1$. The random noise signal *y* is a sequence of length *N* uniformly distributed between $-\sqrt{3}$ and $\sqrt{3}$. Replaced *N × p* points in periodic signal *x* with random noise signal *y*, where *p* represents the probability parameter range from 0 to 1. The data complexity generated by the *MIX* model increases with the increase in *p*.

Extended the *MIX* model to obtain coupling *MIX* model. The coupling *MIX* model consists of the same *MIX* model mixed with two different *MIX* models to generate two-channel signals A and B. The expression of the coupling *MIX* model is as follows:(28)$$\left\{\begin{array}{c} A=C\times MIX(p_{0})+(1-C)\times MIX(p_{1})\\ B=C\times MIX(p_{0})+(1-C)\times MIX(p_{2}) \end{array}\right.$$

The value of probability parameters $p_{0},p_{1},p_{2}$ and coupling coefficient C are both from 0 to 1. The complexity of A and B signal increases as the value of $\left(p_{0},p_{1}\right)$ and $\left(p_{0},p_{2}\right)$ increase, respectively. The correlation between A and B is related to the coupling coefficient C. When C=0, the two signals are not correlated, and when C=1, the data of the two channels are the same. The model parameters are as follows: the coupling coefficient C ranges from 0 to 1 with the step size of 0.1, probability parameters $p_{0},p_{1},p_{2}$ are from 0 to 1, and the data length is N=6000. With the increase in C, the correlation between channels increases gradually. With the change of $p_{0},p_{1},p_{2}$, the complexity of model data will change. Figure 4 shows the variation of multi-channel entropy with coupling coefficient C and single-channel entropy (SC-E).

The figure shows that the multi-channel entropy of mvMSE, mvMFE, and RCmvMFE increases with the coupling coefficient *C* and its single-channel entropy. It shows that the entropy obtained by these three algorithms is affected by different signal complexity and coupling strength. 

For the mvMDE algorithm, the multi-channel entropy increases with the increase in the single-channel entropy when the coupling coefficient C is the same. Moreover, the multi-channel entropy value increases with C when the coupling coefficient C<0.7 and decreases with the rise in C when the coupling coefficient C>0.7 in the case of signal single-channel entropy value is the same. It shows that signal complexity and coupling strength also affect the entropy obtained by the mvMDE algorithm. However, it cannot reflect the complexity and coupling strength of the signal accurately.

For mvMPE and mvMIE algorithms, the multi-channel entropy increases with the increase in single-channel entropy when the coupling coefficient C is the same. However, when the single-channel entropy of the signal is the same, the multi-channel entropy is unchanged with the increase in the coupling coefficient C. It indicates that the entropy obtained by these two algorithms cannot use to measure the coupling strength of signals with the same complexity.

As can be seen from this simulation analysis: mvMPE and mvMIE algorithms cannot comprehensively describe the coupling strength between signal channels. The mvMDE algorithm is inaccurate when measuring the coupling strength between signal channels. However, the three algorithms of mvMSE, mvMFE, and RCmvMFE can better reflect the influence of the coupling strength between signal channels and the complexity of single-channel. 

#### 3.1.4. Noise Resistance Analysis

To analyze the anti-noise performance of the six algorithms, we superimposed white noise with a different signal-to-noise ratio (SNR) on the coupled *MIX* model data to generate simulation signals. The model parameters are as follows: C=0.4, $p_{0}=0.2$, $p_{1}=0.5$, $p_{2}=0.8$ [37]. The data length is 6000 and repeats the generation 20 times. Then, add 20 dB, 10 dB, 0 dB, −10 dB, and −20 dB WGN on the model data to generate simulation signals with different SNRs. Calculate the entropy values of the simulation signals by six algorithms and compute their mean and standard deviation. The results are shown in Figure 5.

The figure shows that when the additive noise is 20 dB and 10 dB noise to the original signal, the six multivariate multi-scale entropy algorithms show good performance in anti-noise. When the additive noise is 0 dB, except for the mvMDE algorithm, the entropy values change curves obtained by the other five multivariate multi-scale entropy algorithms are between the entropy change curves of 10 dB noise and −10 dB noise. 

However, when the additive noise is −10 dB and −20 dB, the results of mvMSE, mvMDE, mvMFE, RCmvMFE, and mvMPE algorithms are too different from that of the original signal. In the four algorithms of mvMSE, mvMDE, mvMFE, and RCmvMFE, the entropy values of the four algorithms with additive noise of −10 dB and −20 dB show an overall downward trend at any scale factors, which is very different from the entropy values of the original signal. The entropy result is larger than the original signal entropy value when the scale factor is less than 4. When the scale factor is greater than 6, the entropy result decreases with the increase in noise intensity and is less than the original signal entropy value. The deviation between the entropy value and the original signal entropy value increases with the growth of the scale factor, showing a big difference from the original signal entropy change trend. 

For the mvMPE algorithm, the entropy value is mainly unchanged with the increase in the scale factor with additive noise of −10 dB and −20 dB. When the scale factor is less than 4, the entropy value is almost equal to the original signal entropy value. When the scaling factor is greater than 5, the entropy value still changes little, and the characteristic that the original signal entropy value varies with the increase in the scale factor is lost. For the mvMIE algorithm, when the additive noise is −10 dB and −20 dB and the scale factor is greater than 2, the entropy value is lower than the original signal entropy value and unchanged with the increase in the scale factor. When the additive noise is −20 dB and −10 dB, the original signal submerges by the noise, and the entropy curve of the original signal is lost. When we analyze the actual signals, we should remove the noise interference as much as possible to avoid a large deviation in the results.

#### 3.1.5. Data Length Analysis

Use the coupling *MIX* model for simulation analysis to explore the influence of data length on multivariate multi-scale entropy algorithms. The model parameters are set as follows: C=0.4, $p_{0}=0.2$, $p_{1}=0.5$, $p_{2}=0.8$. The data length N ranges from 100 to 2000 with a step size of 100 and repeats the generation 20 times. Calculate the entropy values of the generated data by six algorithms and compute their mean and standard deviation. The results are shown in Figure 6.

As seen from the figure, with the continuous increase in data length N, the values of the six multivariate multi-scale entropy algorithms all show an upward trend and gradually tend to be stable. The entropy values of mvMSE, mvMPE, and mvMIE algorithms tend to be stable when N=700, while mvMDE, mvMFE, and RCmvMFE algorithms are basically stable when N=500.

Based on all results of the above simulation analysis, the RCmvMFE algorithm has the best comprehensive performance ability. This algorithm is affected by signal complexity and correlation between signals and decreases with the decrease in signal complexity and coupling strength. According to existing research, complexity and functional connectivity coexist in the actual EEG, especially in the MCI EEG signal, which showed a trend of decreased complexity and functional connectivity compared with healthy controls [16,33,34]. After neurofeedback training, MCI patients have improved cognitive function and significantly increased signal complexity and functional connectivity across the brain simultaneously [38,39]. Combined with the characteristics of MCI EEG signals, we believe that the RCmvMFE algorithm is the most suitable algorithm for analyzing MCI EEG data among the six algorithms and is more likely to detect changes in the complexity and functional connectivity of MCI EEG data.

### 3.2. Real EEG Data Analysis

#### 3.2.1. The RCmvMFE Analysis of EEG

Divide the EEG signals of 180 s into non-overlapping five-second data segments. Use the RCmvMFE algorithm to analyze the data segments of different brain regions. The mean value obtained after removing abnormal values is the multi-channel entropy of each brain region. Set the parameters of the RCmvMFE algorithm as follows: the embedding dimension of each channel is 2; the time delay is 1; the threshold value r=0.15×std; the adjustment factor λ=0.8; and the scale factor ranges from 1 to 20. Each segment of data is 5 s without overlap. The curves of RCmvMFE values of EEG signals in MCI and control groups of each brain region are shown in Figure 7. It can see from the figure that, in the six brain regions, when the scale factor is less than or equal to 4, the entropy of the MCI group is generally less than that of the control group. When the scale factor is greater than or equal to 10, the entropy of the MCI group is greater than that of the control group. The entropy changes of EEG signal in MCI patients under different scale factors may be related to cognitive decline. The decrease in entropy of EEG signals in MCI patients under small-scale factors may be associated with the synaptic inefficiency caused by neuron death, neurotransmitter changes, and loss of local neural network connection. Therefore, a more extensive range of neurons is required to participate in neural activities, increasing entropy under large-scale factors.

In addition, we compared the RCmvMFE values in each brain region between the aMCI group and control group at the short scale (1≤s≤4) and the long scale (10≤s≤15), respectively. The RCmvMFE values of each brain region in the MCI and control groups were consistent with normality and homogeneity of variance on the short and long scales and statistically analyzed by independent sample *t*-test in statistical analysis software SPSS (version 25.0). Adopt the false discovery rate (FDR) and set the correction level FDR≤0.05. The detailed information on RCmvMFE values and statistical analysis results in each brain region between the two groups are shown in Figure 8. From this figure, we can more intuitively observe the differences between MCI and control groups in each brain region on the short and long scales. On the short scale, there are significant differences in frontal, central, parietal, and occipital, and the difference in frontal was significantly more (p<0.001); on the long scale, there were significant differences in central, parietal, and left temporal.

#### 3.2.2. Correlation between EEG Entropy and Cognitive Function

We calculated the Pearson linear correlation coefficient between the RCmvMFE values of EEG signals and the neuropsychological scale scores of all subjects to study the relationship between the entropy values of each brain region and cognitive function. Use the FDR method for strict correction and set the correction level as FDR≤0.05. We obtained scatter diagrams describing the significant correlation between entropy values and neuropsychological scale scores, as shown in Figure 9. On the short scale, the RCmvMFE values in frontal have a significant positive correlation with the test scores of MoCA and AVLT delayed recall.

## 4. Discussion

Currently, many EEG analysis algorithms use to study EEG signals in functional brain diseases. Among them, using entropy to process nonlinear signals has attracted extensive attention. In this paper, we analyze six algorithms from a new perspective to solve the problem that other reports do not specify what characteristics of signals the algorithm measures and to explain which algorithm is more suitable for analyzing MCI EEG signals. It is worth noting that we explore the relationship between multi-channel entropy, single-channel entropy, and inter-channel coupling for the first time, which is new and most important in our simulation research and is unprecedented in previous papers.

From the simulation analysis results, we found that when the SNR was 20 dB, the entropy curve was close to the original signal curve, and then with the increase in SNR, the entropy curve obtained by the six algorithms all deviated to some extent. Among them, the mvMIE algorithm is the least affected by noise, which may be related to the *MIX* coupling model data since the algorithm is the entropy calculation of the signal after pairwise difference. The mvMDE algorithm can analyze the intra-channel correlation and multivariable signal complexity and identify whether there is a correlation between 1/f noise signal channels with an inaccuracy measurement. The mvMPE and mvMIE algorithms cannot reflect the correlation within and between channels, so they cannot describe the signal comprehensively. The mvMSE, mvMFE, and RCmvMFE algorithms can not only measure intra-channel correlation and multivariable signal complexity but also distinguish whether there is a correlation between signal channels. They can well reflect the influence of coupling strength between signal channels and single-channel complexity. The RCmvMFE algorithm has the best performance, which may be related to the concept of the algorithm. The mvMSE algorithm combines multivariate sample entropy and multi-scale entropy to make it possible to analyze multi-channel signals with different scale factors. It can analyze multi-channel data’s long-term correlation and complexity well and receives wide attention in extracting multi-channel nonlinear signals. The mvMFE algorithm introduces a fuzzy membership function based on the mvMSE algorithm, which improves the system stability of the algorithm and has good consistency. However, considering the data length of different scale factors varies greatly, information loss will inevitably occur. The RCmvMFE algorithm primely solves this problem by improving the coarse-grained algorithm and can stably detect the dynamic behavior of complex systems. In summary, the RCmvMFE algorithm has the most significant description effect and is more likely to detect changes in MCI EEG signal complexity and functional connectivity.

In this paper, we used the RCmvMFE algorithm to calculate the entropy of the EEG signal in each brain region of 20 subjects (8 control subjects and 12 MCI subjects). We observed the changing trend of RCmvMFE entropy values and statistically analyzed whether there was a significant difference between the two groups. On the short scale, the entropy of the MCI group was lower than the control group, and there were significant differences in the frontal, central, parietal, and occipital regions, especially the frontal region. On the long scale, the entropy of the MCI group was higher than that of the control group, with significant differences in the central, parietal, and left temporal regions. These results are consistent with previous studies showing that the short-scale entropy of EEG in cognitive disorder patients (MCI or AD) is lower than that in healthy people, and the long-scale entropy is higher than that in healthy people [40,41,42,43,44]. That may be related to the inefficiency of local neural network connection caused by neuron death and neurotransmitter changes, which requires a broader range of neurons to participate in neural activity. After that, we calculated the Pearson correlation coefficient between the entropy values of all brain regions and neuropsychological scale scores. After FDR strict correction, we found that on the short scale, the frontal entropy has a significant positive correlation with the MoCA score and AVLT delayed recall score, indicating that the damage to cognitive function in patients with MCI may be related to the frontal region.

The above studies indicate that the RCmvMFE value is the EEG characteristics related to cognitive impairment, which can be used as EEG markers for the early diagnosis of MCI and provide help for the diagnosis and treatment of MCI. This algorithm is very suitable for analyzing MCI EEG signals. However, this algorithm may have some limitations when applied to EEG analysis of other brain function diseases. It is necessary to know the changing characteristics of EEG signals in this disease and whether the trend affected by signal complexity and functional connectivity is the same. If the effects are opposite, this algorithm is not suitable for analyzing the EEG of this disease.

Some deficiencies in our research content still need to be improved, such as a small number of EEG channels analyzed, a small number of research subjects studied, and a lack of sensitivity and specificity analysis of algorithm classification. In the future, other researchers or we can collect EEG signals from more subjects and add AD patients to analyze the differences in EEG characteristics between the control, mild cognitive impairment, and AD groups. The proposed algorithm can combine with various classifier methods to classify the subjects according to the signal features extracted by the algorithm. These will be the content of our further research in the future.

## 5. Conclusions

This paper explicitly discusses what signal characteristics algorithms measure and which factors affect them for the first time, explains which algorithm is more suitable for analyzing the MCI EEG signal, then analyzes the actual EEG signal. Among them, the analysis of the relationship between multi-channel entropy, single-channel entropy, and inter-channel coupling is the most important novel point, which has been undiscovered in previous papers.

Simulation results show that the mvMSE, mvMFE, and RCmvMFE algorithms can analyze intra-channel correlation and multivariable signal complexity and accurately measure the coupling strength when inter-channel correlation occurs. The mvMSE, mvMFE, and RCmvMFE algorithms can well reflect the influence of coupling strength between signal channels and single-channel complexity, among which the RCmvMFE algorithm is the most prominent. The mvMDE algorithm can analyze the intra-channel correlation and multivariable signal complexity, whereas it cannot accurately measure the coupling strength when the inter-channel correlation. The mvMPE and mvMIE algorithms cannot measure the intra-channel and inter-channel correlation and can only use to analyze the multivariable signal complexity with poor discrimination ability. So, the mvMPE and mvMIE algorithms cannot comprehensively describe the signals’ characteristics.

Based on the simulation results, the RCmvMFE algorithm has the best performance in measuring the intra-channel correlation, inter-channel correlation, and multi-channel signal complexity, which is more suitable for analyzing MCI EEG signals. Therefore, we used the RCmvMFE algorithm to analyze the actual EEG signals of the MCI group and the control group. The results showed that on the short scale (1≤s≤4), the entropy values of the MCI group were lower than the control group, and there were significant differences in the frontal, central, parietal, and occipital regions, among which the difference in the frontal was significantly most (p<0.001). On the long scale (10≤s≤15), the entropy values of the MCI group were higher than the control group, and significant differences in the central, parietal, and left temporal regions. The entropy value obtained by RCmvMFE in this paper was the EEG characteristics related to cognitive impairment, which was considered a potential biomarker for the diagnosis of MCI and provided help for the early diagnosis and treatment of MCI.

## Figures and Tables

**Figure 1 entropy-25-00396-f001:**
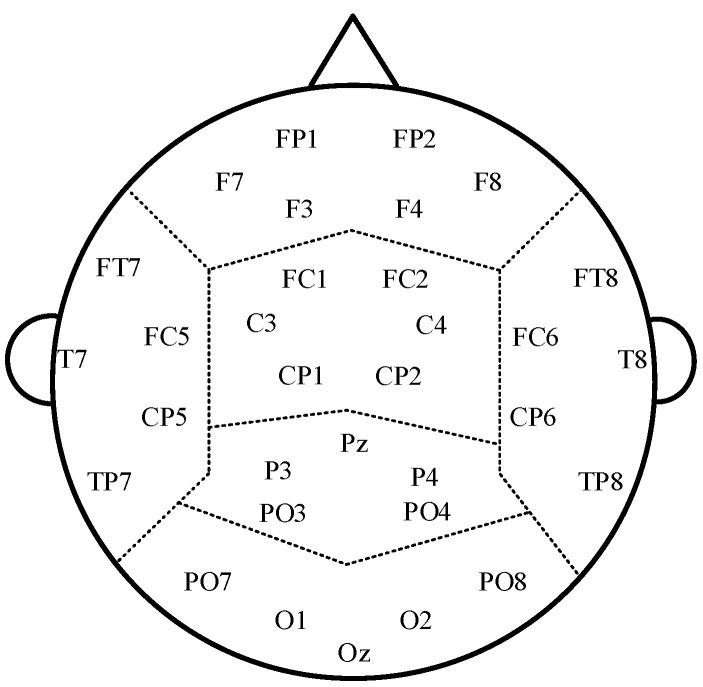
The interesting channels and brain regions.

**Figure 2 entropy-25-00396-f002:**
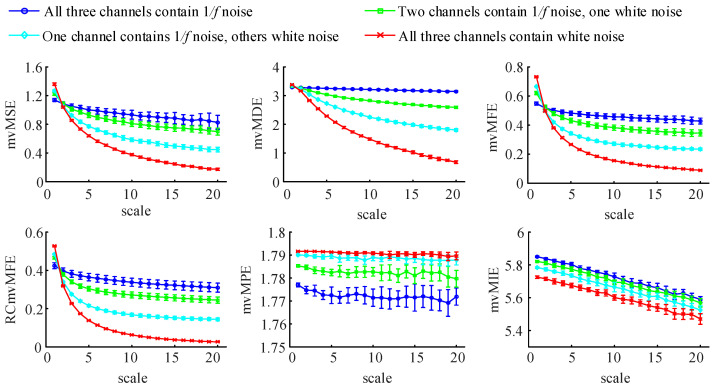
The mean and standard deviation of the results using six multivariate multi-scale entropy algorithms compute from 20 different uncorrelated three-channel time series with 6000 data points.

**Figure 3 entropy-25-00396-f003:**
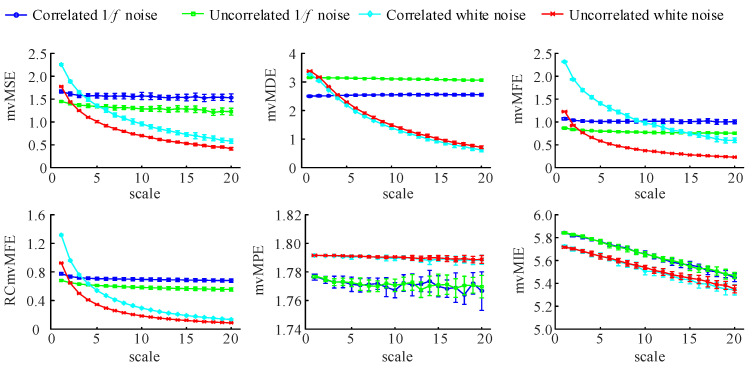
The mean and standard deviation of the results using six multivariate multi-scale entropy algorithms computed from 20 different correlated and uncorrelated two-channel series with 6000 data points.

**Figure 4 entropy-25-00396-f004:**
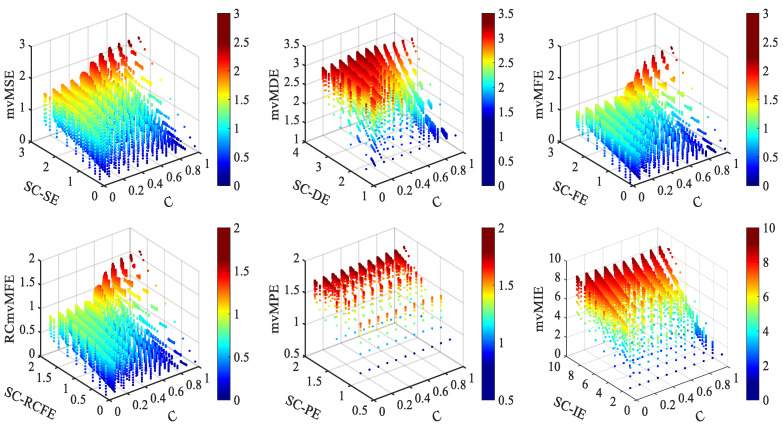
The influence of the coupling strength among signals and single-channel complexity on the entropy value of multi-channel signals calculate by six multivariate multi-scale entropy algorithms. The model data change with coupling coefficient from 0 to 1 with 0.1 steps.

**Figure 5 entropy-25-00396-f005:**
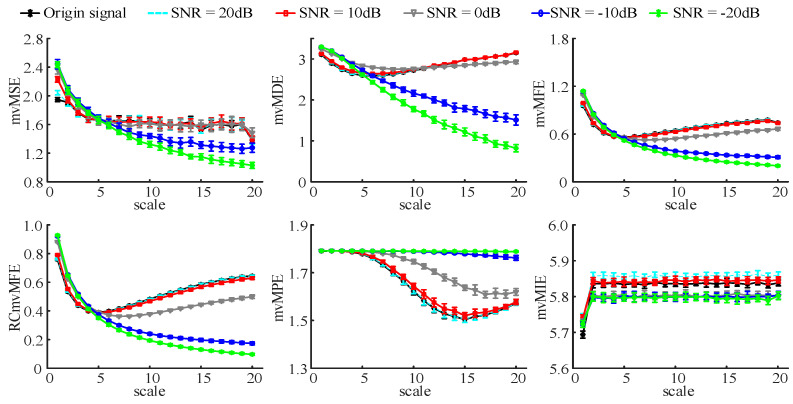
The mean and standard deviation of the results using six multivariate multi-scale entropy algorithms compute from 20 different noise model data with 6000 data points.

**Figure 6 entropy-25-00396-f006:**
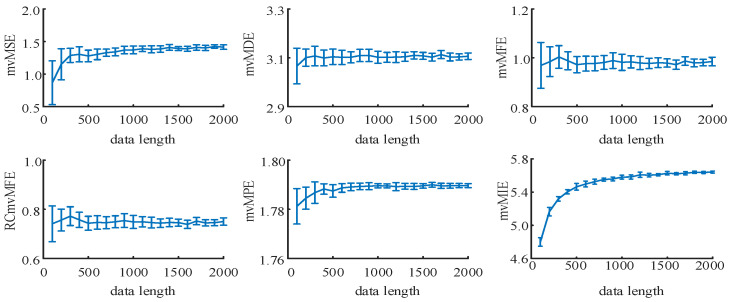
The mean and standard deviation of the results using six multivariate multi-scale entropy algorithms compute from 20 different model data lengths ranging from 100 to 2000 with a step of 100.

**Figure 7 entropy-25-00396-f007:**
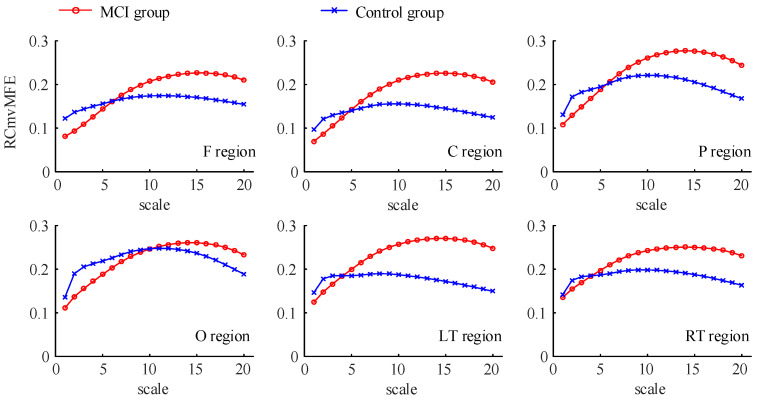
RCmvMFE values of brain regions in the MCI and control group.

**Figure 8 entropy-25-00396-f008:**
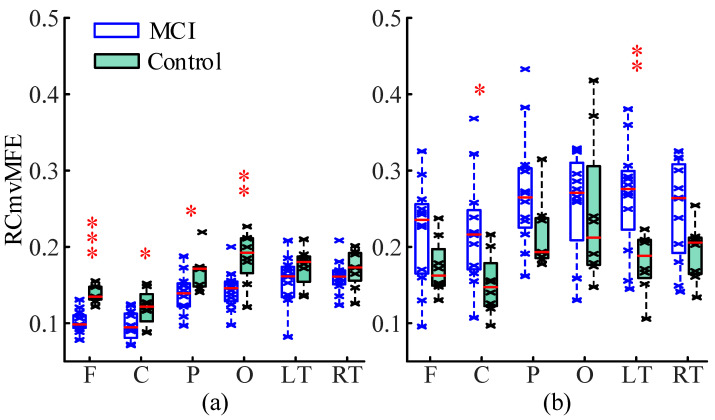
The detailed information on RCmvMFE values and statistical analysis results in each brain region in the MCI and control group (**a**) on the short scale and (**b**) on the long scale. The ‘x’ symbol represents the RCmvMFE value, the red horizontal line represents the median, and the ‘*’ sign marks a significant difference after FDR correction. The ‘x’ symbol represents the RCmvMFE value, the red horizontal line represents the median, and the ‘*’ sign means a significant difference after FDR correction, where ‘*’ indicates 0.01 ≤ *p* < 0.05, ‘**’ indicates 0.001 ≤ *p* < 0.01, and ‘***’ indicates *p* < 0.001.

**Figure 9 entropy-25-00396-f009:**
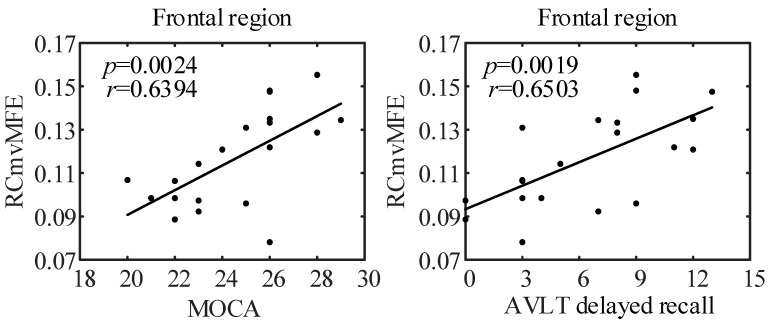
Correlation analysis between the RCmvMFE and cognitive function.

**Table 1 entropy-25-00396-t001:** Demographic and neurological scales statistical analysis of all subjects.

Factors	MCI Group	Control Group	*p*-Value
Subject count	12	8	--
Age	71.083 ± 6.828	71.375 ± 7.539	0.929
MoCA	23.000 ± 1.758	26.875 ± 1.246	*p* < 0.001 ***
MMSE	28.416 ± 1.928	28.750 ± 0.462	0.576
BOSTON naming	19.333 ± 0.887	19.750 ± 0.462	0.188
AVLT immediate recall	5.891 ± 1.693	7.850 ± 1.619	0.019 *
AVLT delayed recall	4.333 ± 3.498	9.625 ± 2.133	0.001 **
AVLT long-delayed recognition	11.000 ± 4.177	13.750 ± 1.388	0.091
Semantic fluency test	18.250 ± 4.092	18.625 ± 3.248	0.831
Trail test A	62.818 ± 18.296	62.125 ± 19.445	0.938
Trail test B	116.454 ± 38.388	105.125 ± 40.814	0.544
WAIS	11.000 ± 2.898	13.875 ± 1.125	0.017 *
FAQ	2.723 ± 3.849	0.625 ± 1.767	0.171

* Indicates 0.01≤ *p* < 0.05, ** indicates 0.001≤ *p* < 0.01, and *** indicates *p* < 0.001.

## Data Availability

The data supporting the findings of this study are available from the Special Medical Center of the Chinese People’s Liberation Army Rocket Force, but restrictions apply to the availability of these data, which requires permission and so are not publicly available. However, data are available from the authors upon reasonable request and with permission of the Special Medical Center of the Chinese People’s Liberation Army Rocket Force.

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
