# Peer review of "Which Multivariate Multi-Scale Entropy Algorithm Is More Suitable for Analyzing the EEG Characteristics of Mild Cognitive Impairment?"

_entropy, 2023, doi:10.3390/e25030396_

Round 1

Reviewer 1 Report

This article analyzes six commonly used multidimensional multiscale entropy algorithms and the characteristics of signals depending on the factors that affect them. In particular, the electroencephalogram (EEG) signals of mild cognitive impairment (MCI) are considered. The algorithms of multivariate multi-scale sample entropy (mvMSE), multivariate multi-scale fuzzy entropy (mvMFE), and refined composite multivariate multi-scale fuzzy entropy (RCmvMFE) can measure intra-channel correlation and multivariate signal complexity and accurately measure inter-channel correlation. Among them, the RCmvMFE algorithm has the best result.
I'd like to ask about next aspects of the presented result:
could this algorithm be used for crisp and fuzzy classifiers?
what classifiers can be recommended?
could you consider the application of this entropy is the methods for EEG signal classification proposed in https://ieeexplore.ieee.org/document/9224666
Does the proposed algorithm efficiency depend on another sample of EEG signal and can it be used in other EEG analysis based problems?

Reviewer 2 Report

The introduction of the various complexity metrics starting at line 58 could be made more explanatory, as currently it reads as a quick list of bulletpoints.

There are many strange phraseologies, e.g.:
- line 136: "The paper is arranged as follows..."

There is a general lack of explanation. It is clear that the authors are really immersed in the topic, but communicating the work to people outside the research group is equally important. The entire paper reads as a powerpoint presentation given internally to the research group. The entire methods section is not friendly to the reader. A more reader-friendly version would provide examples. For instance, in line 57 is a good opportunity to show an example or how entropy is used in at least two fields. These examples would use different metrics, allowing you to then continue with the large section on different entropy measures. The method section starts immediately with the first metric, mvMSE. Instead, the section could start with a few sentences in which the 6 metrics are named and their main differences highlighted.

The paragraph starting at line 576 should be part of the introduction.

It is clear that the algorithms are compared with each other and an evaluation is given. However, how is "good" (line 296) defined exactly? This extends to section 3.1.2 and 3.1.3. There is a quantitative simulation study, but the evaluation of good or bad is made subjectively. How much better is RCmvMFE against the others?

The application to real EEG data is good. It would be interesting to know whether the other complexity measures distinguish MCI for healthy controls and compare all metrics in their sensitivity and specificity in classification. Ideally, ROCs would also be presented.

Finally, given the importance of the scale parameter both in the simulations and in the analysis of real EEG data, is the pattern due to a mathematical tradeoff or is there a real difference in entropy in the EEG data. For example, does the MCI data contain more "white noise" than the Control data? More information about why the data results appear as they do is needed.

Reviewer 3 Report

This article presents an analysis of six commonly used multivariate multi-scale entropy algorithms to find the EEG characteristics of Mild Cognitive Impairment. In a first step the analysis was made in synthetic signals and in a second step in EEG real signals.

This work shows an interesting investigation to characterize the MCI and its relation with the T2DM. The manuscript is well presented, and the methodology is clear. Nevertheless, some considerations are suggested before publication.

- For the preprocessing EEG step, what method was used to artifacts remove? Which is the Wavelet enhanced independent component analysis?

- The authors mention that the EMG, EEG, and ECG artifacts were removed. It is clear that the EMG and ECG artifacts are the signals of the muscular and cardiac activity, but what are the EEG artifacts? The cerebral activity was removed?

- What is the advantage of down-sampling the EEG signals in this analysis?

- The authors mentioned that thirty-two interesting channels were selected. How many electrodes be registered originally?

- In the simulation section, the figures presented different scales between the six subfigures of each one. This difference does not allow to compare the methods properly. Also, Figure 8 of the real signal analysis has different scales.

- The 20 EEG recordings are not couplets. It would be better to present the same number of control and  MCI subjects. These differences between the subject groups can show unwanted trends in the results.

-Finally I suggest the authors to remark clearly the novelty of the presented work.

Reviewer 4 Report

The manuscript investigates six commonly used multivariate multiscale entropy algorithms in an EEG dataset of MCI patients and healthy controls. The work provides an in-depth introduction. The experimental results (adopting the RCmvMFE algorithm) show that the entropy values of the MCI group were lower than the control group.

Overall, I think the paper needs to undergo minor revisions before publication could be considered. My comments, questions and suggestions are listed below.

-The paper looks good other than a few spelling/grammar typos and one formatting issue.

- Why have authors used FDR correction? Why not Bonferroni?

- The authors may briefly discuss the potential limitations of the proposed method and what are the future research directions of this study. How other researchers can work on your study to continue this line of research?

Round 2

Reviewer 1 Report

The comments in the enclosed file

Reviewer 3 Report

I don't have any comments.

Round 3

Reviewer 1 Report

I have no comments

Author Response

Thank you so much for approving this article.